# Biological Activities and Chemical Composition of *Santolina africana* Jord. et Fourr. Aerial Part Essential Oil from Algeria: Occurrence of Polyacetylene Derivatives

**DOI:** 10.3390/molecules24010204

**Published:** 2019-01-08

**Authors:** Charaf Eddine Watheq Malti, Clémentine Baccati, Magali Mariani, Faiçal Hassani, Brahim Babali, Fewzia Atik-Bekkara, Mathieu Paoli, Jacques Maury, Félix Tomi, Chahrazed Bekhechi

**Affiliations:** 1Laboratoire des Produits Naturels, Département de Biologie, Université Abou Bekr Belkaïd, Imama Tlemcen 13000, Algeria; mcew.malti@gmail.com (C.E.W.M.); f_atik@yahoo.fr (F.A.-B.); bekhechichahrazed@yahoo.fr (C.B.); 2Université de Corse-CNRS, UMR 6134 SPE, Route des Sanguinaires, 20000 Ajaccio, France; clementine.baccati@gmail.com (C.B.); mariani_m@univ-corse.fr (M.M.); mathieu.paoli@univ-corse.fr (M.P.); maury_j@univ-corse.fr (J.M.); 3Laboratoire d’Ecologie et Gestion des Ecosystèmes Naturels, Département d’Ecologie et Environnement, Université Abou Bekr Belkaïd, Imama Tlemcen 13000, Algeria; faicalhassani@yahoo.fr (F.H.); miharb_babali@hotmail.fr (B.B)

**Keywords:** *Santolina africana* Jord. et Fourr., Asteraceae, essential oil composition, ^13^C-NMR, antimicrobial activity, antioxidant activity, anti-inflammatory activity

## Abstract

The chemical composition of 18 oil samples of *Santolina africana* isolated from aerial parts at full flowering, collected in three locations in eastern Algeria was determined by GC(RI), GC/MS and ^13^C-NMR analysis. The major components were: germacrene D, myrcene, spathulenol, α-bisabolol, β-pinene, 1,8-cineole, *cis*-chrysanthenol, capillene, santolina alcohol, camphor, terpinen-4-ol and lyratol. The chemical composition appeared homogeneous and characterized by the occurrence of four derivatives which exhibited a conjugated alkene dialkyne moiety. They were identified for the first time in an essential oil from *S. africana.* The collective oil sample exhibited moderate antimicrobial and antioxidant activities whereas the anti-inflammatory activity presented a real potential. IC_50_ value of *Santolina africana* essential oil (0.065 ± 0.004 mg/mL) is 5-fold higher than IC_50_ value of NDGA used as positive control.

## 1. Introduction

The *Santolina* genus belongs to the Asteraceae family and is represented by more than 10 species widely distributed in Mediterranean area [1]. In all *Santolina* species, *Santolina viridis* W. (South of France and North of Spain), *S. pectinata* Lag. (= *S. rosmarinifolia* L.) (Iberian Peninsula) and *S. chamaecyparissus* L. are the most widely spread species around the world. *S. africana* Jord. et Fourr. is synonym of *Ormenis africana* (Jord. et Fourr.) Litard. et Maire and *S. chamaecyparissus* L. var. *africana* B. et T. It is an endemic species to the North Africa (Morocco, Algeria and Tunisia) [2,3] that grows naturally in forests and steppe pastures. This species is a bushy, green or ashy sub-shrub. The stems are woody, with floriferous branches erect in tuft, bare and thickened at the apex. The lower leaves are linear-cylindrical with short and obtuse segments. The bracts are ovate-oblong. The outer corollas are tube-styled ovary. The flowerheads are discoidal, yellow, homogamous [2].

Some members of *Santolina* genus have been known as medicinal plants for a long time. *S. africana* is used in Moroccan folk medicine as a stomachic, abortive, anthelmintic, antidiabetic and emmenagogue [4,5]. In Tunisia, it is traditionally used for its hypoglycemic effect and for the treatment of stomache pains [6]. Some biological activities have been reported for the extracts or essential oils of *S. africana*, such as antioxidant activity [6,7,8], antimicrobial activity [7], accaricidal activity [9] and antidiabetic activity [8].

The chemical composition of the essential oils of species belonging to the genus *Santolina* has been widely studied [10]. *S. chamaecyparissus* is probably the most investigated species of this genus [11,12,13,14,15,16,17,18,19,20,21,22]. The composition of other species such as *S. corsica* [23,24], *S. insularis* [25,26], *S. rosmarinifolia* [27,28,29] was also reported. Monoterpenes such as 1,8-cineole, camphor, artemisia ketone and myrcene were the major components of essential oils isolated from some *Santolina* species growing in different regions of the word.

Conversely, only five studies have reported on the chemical composition of *S. africana* oil. The chemical composition of the volatile compounds isolated from various parts of the plant have been substantially investigated. Fdil et al. [4] compared the chemical composition of three oil samples isolated from different organs (stems, leaves and flowers) of *S. africana* plants harvested in Marrakech province (Morocco). The three samples oils exhibited respectively a composition dominated by oxygenated monoterpenes: camphor (69.14%/71.36%/80.44%), borneol (20.33%/18.13%/12.34%) and bornyl acetate (7.08%/8.12%/3.50%). The stem oil contained also noticeable amounts of α-humulene (3.14%) while the flowers oil exhibited an appreciable content of 1,8-cineole (3.32%). Another aerial part oil sample of Moroccan origin exhibited a similar composition, with camphor (54.3%), borneol (17.24%), bornyl acetate (8.61%) and 1,8-cineole (5.27%) as main components [5]. A Tunisian oil sample (stems and leaves) was characterized by a high content of terpinen-4-ol (54.96%), followed by α-terpineol (14.06%) and borneol (8.37%) [9]. Concerning Algerian *S. africana*, only two studies are reported in the literature. An oil sample (flowers) harvested in Constantine province (Algeria) was dominated by acenaphtane (25.23%), calarene (21.54%) and ocimene (17.44%) [7]. A drastically different composition has been reported for an aerial parts oil sample collected in the same region, β-eudesmol (14.58%) and β-pinene (12.78%) being the major compounds, followed by 1,8-cineole (10.02%), curcumene (7.96%), myrcene (6.94%) and spathulenol (5.96%) [18].

It appears from literature data that the essential oil (EO) from aerial parts of *S. africana* exhibits a tremendous chemical variability. Moreover, most of the papers reported on the analyses of only one or two oil samples and obviously, the reported compositions are not always representative of *S. africana*. Thus, the aim of the present work is to characterize the EO produced by this plant growing wild in Batna province (Algeria). Eighteen oil samples isolated from aerial parts at full flowering stage harvested in three locations have been analyzed by GC, GC/MS and ^13^C-NMR. Then, the biological activities of the EO have been investigated as antimicrobial, antioxidant and anti-inflammatory activities, this latter has never been investigated in any previous papers.

## 2. Results

Aerial parts of wild *S. africana* were collected in May in three locations (Figure 1) (six samples per location). Yields of EO isolated by hydrodistillation, calculated *w*/*w* vs. dry material varied drastically from sample to sample ranging from 0.03 to 0.17% even within a location (Appendix A). As it could be seen from Appendix A, the highest yields were obtained from Hamla (0.08–0.14%, samples H1–6) and Bouilef (0.15%, sample B5 and 0.17%, sample B6) and the lower from Bouilef and Fesdis (0.03% for samples B2, B4 and F6).

### 2.1. Chemical Composition of Essential Oil

Aerial parts oils samples were submitted to GC-FID analysis, to determine the retention indices (RIs) of the EO components on two columns of different polarity and to GC/MS analysis. Further analysis by ^13^C-NMR confirmed the identification of the main components. To allow the identification of four polyacetylene derivatives present at moderate or low levels, a composite sample (F1 to F6) was submitted to column chromatography (CC) over silica gel. Nine fractions were obtained and analyzed by GC-FID, GC/MS and ^13^C-NMR. In total, 91 components accounting for 92.4% and 96.1% of the whole oil chemical composition were identified (Appendix A, Figure 2), including forty-three monoterpenes, thirty-one sesquiterpenes, five phenylpropanoids, six polyacetylene derivatives and six others. The composition of *S. africana* EOs is generally homogeneous; the oils were found to possess little differences in the chemical composition but considerable variation in the levels of the individual components. All the samples were characterized by high proportions of monoterpenes (51.5–69.7%), except three samples: B4, B5 and F6 which were dominated by sesquiterpenes (44.3–55.9%). The main components were germacrene D (0.1–25.3%), myrcene (4.2–20.9%), spathulenol (2.5–20.7%), α-bisabolol (2.2–20.0%), β-pinene (2.4–18.7%), 1,8-cineole (5.0–16.8%), *ci*s-chrysanthenol (0.7–16.5%), capillene (0.1–16.9%), santolina alcohol (0.2–14.0%), camphor (0.2–7.9%), terpinen-4-ol (1.8–7.3%) and lyratol (0.1–6.7%). Other two oxygenated monoterpenes: lyratal (tr-2.7%) and chrysanthenone (tr-4.5%), three sesquiterpene hydrocarbons: α-curcumene (0.3–3.2%), γ-curcumene (0.1–2.6%) and bicyclogermacrene (0.1–6.3%) as well as two oxygenated sesquiterpenes: β-elemol (up to 3.5%) and β-eudesmol (tr-3.0%) were present in appreciable amounts. Then, one sample for each location (B1, F5 and H1) and five other samples which exhibited various compositions (B3, B4, B5, B6 and H6) were presented in Appendix A.

For instance, the content of capillene (a polyacetylene derivative) reached 16.9% in sample H6 vs. (0.2–7.5%) for the other samples of Hamla and vs. (tr-0.4%) for the samples of Fesdis and Bouilef. The samples B3 and B4 were characterized a high amount of oxygenated sesquiterpenes: spathulenol (15.1% and 20.7%, respectively) associated with α-bisabolol (13.2% and 20.0%, respectively). In the last two atypical samples (B5 and B6), a sesquiterpene hydrocarbon (germacrene D) was present as main compound (25.3 and 20.2%. respectively) vs. (0.0–7.5%) for all other samples.

### 2.2. Identification of Polyacetylene Derivatives

In this study, the identification of four compounds proposed by the MS library was achieved. These compounds were presumed to be two pairs of stereoisomers (*m*/*z* = 200 and *m*/*z* = 198) corresponding to spiroacetalenol derivatives. Indeed, polyacetylene compounds are commonly found in the Asteraceae family [30]. The identification of these compounds was achieved by ^13^C-NMR spectroscopy after fractionation (fractions Fr4 and Fr6, see Experimental part) by comparison of their spectral data with those reported in the literature. (*E*)- and (*Z*)-tonghaosu (*m*/*z* = 200) were identified by comparison with data previously described by Chanotiya et al. [31]. (*E*)-2-(2′,4′-Hexadiynylidene)-1,6-dioxaspiro[4.4]-nona-3,7-diene was identified according to literature data [32] (Appendix A). In the ^13^C-NMR spectrum of Fr4 in which the *E*/*Z* ratio was close to 8/1 (46.6%/6.3%), a series of 13 peaks corresponding to the *Z* isomer was observed. It is the first time that the four spiroacetalenol derivatives [30] were identified in an EO from *S. africana*. The contents of (*E*)-2-(2′,4′-hexadiynylidene)-1,6-dioxaspiro[4.4]-nona-3,7-diene and (*E*)-tonghaosu reached 7.3% (B3) and 3.8% (B6), respectively. These compounds were previously reported in EO from *Chrysanthemum coronarium* L. (aerial parts) [33] and in some solvent extracts from *C. leucanthemum* (roots) [34], *C. coronarium* (aerial parts) [35] and *S. chamaecyparissus* (leaves and buds) [36].

### 2.3. Chemical Variability

The 18 samples were submitted to statistical analyses: the principal components analysis (PCA, covariance) (Figure 3, Appendix A), in which the plan defined by the two axes F1 and F2 described 51.05% of the total variance of the population (the two axes F1 and F2 accounted for 31.70% and 19.35%. respectively). It may be noted that the composition of all samples was qualitatively quite similar. Although the compositions of the individual samples varied substantially for various components, it was not possible to distinguish groups within the essential oil samples. Therefore, one main group (16 samples) and differentiated two atypical compositions (B5 and B6, Figure 3) were observed. Indeed, B5 and B6 were discriminated by a high percentage of sesquiterpene hydrocarbons (bicyclogermacrene, (*E*)-α-bisabolene, γ-curcumene) and particularly germacrene D, 25.3% (B5) and 20.2% (B6) vs. 0–7.5% for the other samples. All the samples from the Fesdis location were homogeneous, while the composition of the samples from the Hamla location appeared much less homogeneous. Conversely, it appeared that the Bouilef samples, located between the two others locations (Fesdis and Hamla) were different, so two samples (B3 and B4) were aggregated to those of the Fesdis location, whereas the two samples B1 and B2 were quite similar to the Hamla samples (Figure 3).

### 2.4. Antimicrobial Activity

The antimicrobial activity of the EO of *S. africana* isolated from the aerial part at full flowering was assayed against four bacteria, two yeasts and three filamentous fungi, using the agar disc diffusion method (Table 1).

The oil was considered active when the diameter of inhibition zone was equal to or greater than 13 mm [24]. The agar diffusion method showed that the oil was effective against *Staphylococcus aureus*, the two yeasts and the three filamentous fungi with diameters of inhibition zone ranging from 13.0 mm to 19.7 mm. The most potent activity was demonstrated against *Staphylococcus aureus* and *Aspergillus fumigatus* with inhibition zones of 19.7 mm and 17.5 mm respectively. In contrast, the growth of *Bacillus cereus*, *Klebsiella pneumoniae* and *Escherichia coli* were not inhibited by the EO. This assumption is in accordance with previous studies on the *Santolina* genus. Indeed, the *S. africana* EO which contained acenaphtane (25.23%), calarene (21.54%), ocimene (17.44%) as its major components, exhibited a moderate or low activity against the same microorganisms with diameters of inhibition zone ranging from 6.50 mm to 20.15 mm. *Bacillus subtilis* and *Staphylococcus aureus* were the most susceptible to this EO, with inhibition zones of 20.15 and 19.5 mm, respectively [7].

### 2.5. Antioxidant Activity

The results of the DPPH^•^ free radical scavenging test were presented in Table 2 and Appendix A. The EO of the aerial parts of *S. africana* have a high antioxidant activity at concentrations of 32, 64, 128 and 256 mg/mL, with inhibition percentages ranged between 85.54 ± 2.17% and 100 ± 0.00% and an IC_50_ value of 1.51 ± 0.04 mg/mL. However, the antioxidant potential of *S. africana* EO was found to be low than that of ascorbic acid (positive control), with a percentage inhibition of 100%, at a concentration of 2 mg/mL and a significantly lower IC_50_ value of 0.02 ± 0.0005 mg/mL.

### 2.6. Anti-Inflammatory Activity

The anti-inflammatory potential of *S. africana* EO was evaluated by determining its ability to inhibit lipoxygenases (LOX). Indeed, LOXs are a non-heme iron-containing dioxygenases that convert linoleic, arachidonic and other polyunsaturated fatty acid into biologically active metabolites involved in the inflammatory and immune responses. Several inflammatory processes such as arthritis, bronchial asthma and cancer are associated with an important production of leukotrienes catalysed by LOX pathway from arachidonic acid [37,38,39,40]. The inhibition of the LOX pathway with inhibitors of LOX would prevent the production of leukotrienes and therefore could constitute a therapeutic target for treating of human inflammation-related diseases. Thus, the search for new LOX inhibitors appears us critical because many of which exhibit significant anti-inflammatory activity.

The ability of *S. africana* EO to inhibit soybean lipoxygenase was determined as an indication of potential anti-inflammatory activity. *S. africana* EO exhibited an inhibition of LOX activity (Table 2). The percentage of inhibition increases with the concentration of *S. africana* EO, i.e., 23.4% at 0.015 mg/mL to 57.6% at 0.075 mg/mL of EO. No LOX activity could be detected in the presence of 0.1 mg/mL of *S. africana* EO, suggesting almost complete inhibition of LOX activity. The IC_50_ values (concentration at which 50% of the lipoxygenase was inhibited) were determined for the *S. africana* EO and for the non-competitive inhibitor of lipoxygenase, the nordihydroguaiaretic acid (NDGA) (Table 2), usually used as reference in anti-inflammatory assays [38,39,40]. Data showed that the IC_50_ value of *S. africana* essential oil (0.065 ± 0.004 mg/mL) is 5-fold higher than IC_50_ value of NDGA (0.013 ± 0.003 mg/mL).

## 3. Discussion

The composition of *S. africana* aerial part oils isolated from plants growing wild in eastern Algeria (Batna) was different from those reported for oils from Morocco [4,5], Tunisia [9] and Algeria (Constantine) [7,18], but it should be pointed that a similar composition has been reported for flowerhead oil of *S. chamaecyparissus* from Tunisia, which also contained 1,8-cineole, β-eudesmol (10.49%), terpinen-4-ol (6.97%), spathulenol (5.80%), camphor (5.27%) and germacrene D (5.03%) as major components [20]. Other compounds also occurred as main constituents: δ-cadinene (6.55%) and myrtenol (4.26%) which are present at low amounts in all our samples (tr-0.4%).

We can assume that the moderate or low antimicrobial activity of *S. africana* EO is related to one or various major components: 1,8-cineole (12.8%), germacrene D (7.2%), spathulenol (6.2%), *cis*-chrysanthenol (6.0%), myrcene (5.8%), β-pinene (5.2%), α-bisabolol (5.2%), terpinen-4-ol (3.2%), santolina alcohol (3.1%), lyratol (2.4%), capillene (2.2%), (*E*)-α-bisabolene (2.0%), limonene (1.9%), camphor (1.6%) and β-elemol (1.4%). 1,8-Cineole was previously described as antibacterial against *S. aureus* [41] while myrcene which accounted for 57.2% in the fraction Fr1 [23] was already reported as ineffective against *S. aureus* [23,42]. It has been reported also that β-pinene, α-pinene and germacrene D had slight activity against a panel of microorganisms. Indeed, the essential oil of *Pinus nigra* ssp. *pallasiana* (α-pinene. 42.3%; germacrene D. 30.6%) exhibited a low antimicrobial activity against the same strains with MICs in the range 10–20 mg/mL [43], so it has been demonstrated in the literature that the inhibitory activity of an EO results from a complex interaction between its different constituents, which may produce additive, synergistic or antagonistic effects, even for those present at low concentrations, i.e., 1,8-cineole in combination with camphor has shown higher antimicrobial effects [44]. In parallel, Lemos et al. [45] reported that the essential oil of *Rosmarinus officinalis* which contained camphor (24.4–35.9%) as major compound exhibited a high antimicrobial activity against *S. aureus* with MICs in the range 0.5–2.0 µL/mL. Otherwise, an interesting antimicrobial activity of a lyratol-rich fraction (84%) was observed against *S. aureus* (19 mm), suggesting that lyratol could be the main responsive of the antimicrobial properties of *Santolina corsica* [23]. It has been summarized also that oxygenated terpenes, as well as alcohols which are present in appreciable amounts in our oil, are active but with differing specificity and levels of activity [46,47].

In previous studies, Derouiche et al. [7] reported a percentage inhibition of the free radical DPPH of the *S. africana* flower EO of about 13.80% at a concentration of 0.1 M, a value much lower than ascorbic acid (more than 70% of inhibition) used as a positive control. Nouasri et al. [17] evaluated the antioxidant activity of the essential oil of the aerial parts of *S. chamaecyparissus* using two methods, the DPPH^•^ free radical scavenging test and the β-carotene bleaching test. They reported that *S. chamaecyparissus* EO had low antioxidant capacity to reduce DPPH^•^ radical with an IC_50_ of about 43.01 ± 8.04 mg/mL, compared to BHT (IC_50_ = 0.072 ± 0.001 mg/mL) and ascorbic acid (IC_50_ = 0.004 ± 0.001 mg/mL). The β-carotene bleaching test revealed that the EO had a moderate activity with a percentage inhibition of the oxidation of linoleic acid of the order of 47.00 ± 3.13%, a value that is higher than that of ascorbic acid tested (11.05%), but much lower than BHT (96.92%).

The measurement of antioxidant activity has revealed that aerial parts of *S. africana* EO exhibited an antioxidant activity that could have an eventual possibility to be used in the food industry, as a natural antioxidant agent, for the preservation of foodstuffs, or in the field of health, for the prevention of various diseases.

Concerning the anti-inflammatory activity, the low ratio between the two values of IC_50_ (*S. africana* EO vs. NDGA) makes it possible to consider the *S. africana* EO as a high inhibitor of the LOX activity [48]. Thus, according to the results, *S. africana* EO exhibits a high inhibition of LOX activity, suggesting an anti-inflammatory potential.

## 4. Materials and Methods

### 4.1. Plant Material

Aerial parts of *Santolina africana* were collected during the flowering period in May 2016 in three locations in the Batna province (Eastern Algeria): Fesdis (Fesdis: F1–6; Bouilef: B1–6) and Oued Chaaba (Hamla: H1–6) (Figure 1). Identification of the plant material was performed by Dr. Babali B., (Laboratory of Ecology and Management of Natural Ecosystems, University of Tlemcen, Imama Tlemcen, Algeria). A voucher specimen has been deposited at the Laboratory of Natural Products (Department of Biology, University of Tlemcen, Algeria), under the accession n° A. 2844. The essential oil was obtained by hydrodistillation of dried aerial parts (around 150–280 g) for 2 h. Yields have been calculated from dry material.

### 4.2. Gas Chromatography (GC) Analysis

GC analyses were performed on a Clarus 500 FID gas chromatograph (PerkinElmer, Courtaboeuf, France) equipped two fused silica gel capillary columns (50 m × 0.22 mm, film thickness 0.25 µm), BP-1 (polydimethylsiloxane) and BP-20 (polyethylene glycol). The oven temperature was programmed from 60 to 220 °C at 2 °C/min and then held isothermal at 220 °C for 20 min, injector temperature: 250 °C; detector temperature: 250 °C; carrier gas: hydrogen (1.0 mL/min); split: 1/60. The relative proportions of the oil constituents were expressed as percentages obtained by peak area normalization, without using correcting factors. Retention indices (RIs) were determined relative to the retention times of a series of *n*-alkanes with linear interpolation (‘Target Compounds’ software of PerkinElmer).

### 4.3. Mass Spectrometry

The EOs were analyzed with a PerkinElmer TurboMass detector (quadrupole, PerkinElmer, Courtaboeuf, France), directly coupled to a PerkinElmer Autosystem XL (PerkinElmer), equipped with a fused silica gel capillary column (50 m × 0.22 mm i.d., film thickness 0.25 µm), BP-1 (dimethylpolysiloxane). Carrier gas, helium at 0.8 mL/min; split: 1/75; injection volume: 0.5 µL; injector temperature: 250 °C; oven temperature programmed from 60 to 220 °C at 2 °C/min and then held isothermal (20 min); ion source temperature: 250 °C; energy ionization: 70 eV; electron ionization mass spectra were acquired over the mass range 40–400 Da.

### 4.4. NMR Analysis

^13^C-NMR analyses were performed on an AVANCE 400 Fourier Transform spectrometer (Bruker, Wissembourg, France) operating at 100.623 MHz for ^13^C, equipped with a 5 mm probe, in CDCl_3_, with all shifts referred to internal tetramethylsilane (TMS). ^13^C-NMR spectra were recorded with the following parameters: pulse width (PW): 4 µs (flip angle 45°); acquisition time: 2.73 s for 128 K data table with a spectral width (SW) of 220.000 Hz (220 ppm); CPD mode decoupling; digital resolution 0.183 Hz/pt. The number of accumulated scans ranged 2000–3000 for each sample (around 40 mg of oil in 0.5 mL of CDCl_3_). Exponential line broadening multiplication (1.0 Hz) of the free induction decay was applied before Fourier transformation.

### 4.5. Identification of Individual Components

Identification of the components was based: (i) on comparison of their GC retention indices (RIs) on polar and apolar columns, determined relative to the retention times of a series of *n*-alkanes with linear interpolation (‘Target Compounds’ software of PerkinElmer), with those of authentic compounds and (ii) on comparison of the signals in the ^13^C-NMR spectra of EOs with those of reference spectra compiled in the laboratory spectral library, with the help of a laboratory-made software [49,50,51]. In the investigated samples, individual components were identified by NMR at contents as low as 0.4%. Several compounds were identified by comparison of ^13^C-NMR chemical shifts with those reported in the literature, for instance capillene and capillin [27,52]; (*E*)-2-(2′,4′-hexadiynylidene)-1,6-dioxaspiro[4.4]-nona-3,7-diene [32]; (*Z*) and (*E*)-tonghaosu. (2-(2′,4′-hexa-diynyl-idene)-1,6-dioxaspiro[4.4]-non-3-ene [31,53].

### 4.6. Essential Oil Fractionation

A composite oil sample (F1 to F6, Fesdis; 247.9 mg) was submitted to flash chromatography (silica gel: 35–70 µm). Nine fractions (Fr1-Fr9) were eluted with a mixture of solvents of increasing polarity (pentane:diethyl ether, 100:0 to 0:100, and pure methanol); Fr1 (10.3 mg) and Fr2 (12.6 mg); pentane:Et_2_O, 98:2; Fr3 (9.2 mg); pentane:Et_2_O, 95:5 Fr4 (14.4 mg); pentane:Et_2_O, 90:10 Fr5 (18.8 mg); pentane:Et_2_O, 75:25 Fr6 (86.5 mg); pentane:Et_2_O, 50:50 Fr7 (18.7 mg); pentane:Et_2_O, 0:100; Fr8 (22.3 mg) and pure methanol, Fr9 (15.6 mg). All fractions of chromatography were analyzed by GC (RI), GC/MS and ^13^C-NMR.

### 4.7. Antimicrobial Activity of the Essential Oil

#### 4.7.1. Microbial Strains

Antimicrobial activity of the aerial part EO (Collective sample Bouilef-Hamla) were evaluated against two Gram-positive bacteria (*Staphylococcus aureus* ATCC 6538 and *Bacillus cereus* ATCC 25921) and two Gram-negative bacteria (*Escherichia coli* ATCC 8739, *Klebsiella pneumoniae* ATCC 700603), two yeasts (*Candida albicans* ATCC 26790 and *C. albicans* ATCC 10231) and three filamentous fungi (*Fusarium oxysporum* MNHN 963917, *Aspergillus fumigatus* MNHN 566 and *Aspergillus flavus* MNHN 994294).

#### 4.7.2. Screening of Antimicrobial Activity

The agar diffusion method [54] was used for the determination of antimicrobial activity of the EOs. Briefly, a suspension of the tested microorganisms (1 mL of a suspension at 10^6^ cells/mL for bacteria and yeasts, 10^7^ cells/mL for *S. aureus* and 10^4^ spores/mL for filamentous fungi) was spread on the solid media plates, using Mueller–Hinton agar for bacteria, Sabouraud dextrose for yeasts and PDA for filamentous fungi. Filter paper discs (6 mm in diameter) were impregnated with 15 μL of the oil and 5 μL of DMSO and placed on the surface of inoculated plates. The activity was determined by measuring the inhibitory zone diameter in mm after incubation for 24 h at 37 °C for bacteria, 24–48 h at 30 °C for yeasts and 3 to 5 days at 25 °C for filamentous fungi. Fluconazole (FLU 25 µg/disc), nystatin (NY 30 µg/disc) were used as reference antifungal against yeasts and filamentous fungi and chloramphenicol (CHL 30 μg/disc), ciprofloxacin (CIP 10 μg/disc), gentamicin (GMN 10 µg/disc), vancomycin (VAN 30 µg/disc) were used as positive controls against bacteria. DMSO was used as negative control. Each test was performed in duplicate or in triplicate.

### 4.8. DPPH Radical Scavenging Activity

The antioxidant activity was measured on a sample of EO (Collective sample Fesdis F1–6). The antioxidant activity of *S. africana* EO was measured on the basis to scavenge of the 2.2-diphenyl-1-picrylhydrazil (DPPH^•^) free radical, according to the experimental protocol of Blois [55]. A volume of 2.5 mL with various concentrations (256, 128, 64, 32, 16, 8, 4, 2, 1, 0.5, 0.25, 0.125, 0.0625, 0.03125 and 0.015625 mg/mL) of the EO in absolute ethanol were added to 1 mL of an ethanolic solution of DPPH at 0.03 mg/mL. For each concentration, a blank was prepared. In parallel, a negative control is prepared by mixing 2.5 mL of absolute ethanol with 1 mL of ethanolic solution of DPPH. After incubation in the dark for 30 min at room temperature, the absorbance was measured against a blank at 517 nm. The activity of the EO was compared to ascorbic acid as a positive control. DPPH free radical scavenging activity in percentage (%) was calculated using the following formula:DPPH scavenging activity (%) = [(A_control_ − A_sample_)/A_control_] × 100(1)with: A_control_ is the absorbance of the negative control; A_sample_ is the absorbance of the tested sample.

The concentration of the EO required for the 50% reduction in the initial concentration of DPPH (IC_50_) was calculated from the graph plotted of percentage inhibition against essential oil concentrations.

### 4.9. Anti-Inflammatory Capacity of Santolina africana Essential Oil

The anti-inflammatory capacity of *S. africana* essential oil (collective sample Fesdis F1–6) was evaluated by in vitro lipoxygenase inhibition assay [56,57,58]. The in vitro analysis for lipoxygenase inhibitory activity was performed using Lipoxidase type I–B (Lipoxygenase, LOX, EC 1.13.11.12) from Glycine max (soybean) purchased from Sigma-Aldrich Chimie (Saint-Quentin-Fallavier, France). It was determined by kinetic mode of spectrophotometric determination method, which was performed by recording the rate of change in absorbance at 234 nm. Indeed, the increase of absorbance at 234 nm due to formation of 13-hydroperoxides of linoleic acid (substrate used for LOX inhibition activity assay) [56,57,58].

A stock solution of LOX was prepared by dissolving around 5.7 units/mL of LOX in 0.2 M borate buffer pH 9.0 (1 unit corresponding to 1 µmol of hydroperoxide formed per min). Five concentrations of *S. africana* EO in dimethylsulfoxide (DMSO) were tested as inhibitor solution for LOX inhibition activity assay: 1.5, 2.5, 5.0, 7.5 and 10.0 mg/mL.

The LOX inhibition assays were performed by mixing 10 µL of LOX solution with 10 µL of inhibitor solution in 970 µL of boric acid buffer (0.2 M; pH 9.0) and incubating them briefly at room temperature. The reaction started by addition of 10 µL of substrate solution (Linoleic acid, 25 mM) and the velocity was recorded for 30 s at 234 nm. One assay was measured in absence of inhibitor solution and one assay was measured with DMSO mixed with distilled water (99.85% of DMSO in distilled water) which made it possible to eliminate the inhibition effect of DMSO. No inhibitor effect of DMSO on the LOX activity was detected and the LOX activity measured without inhibitor solution was considered as control (100% enzymatic reaction). All assays were performed on triplicate. The percentage of lipoxygenase inhibition was calculated according to the equation:Inhibition % = (V_0control_ − V_0sample_) × 100/V_0control_(2)V_0control_ is the activity of LOX in absence of inhibitor solution and V_0sample_ is the activity of LOX in presence of inhibitor solution [58]. The IC_50_ was calculated by the concentration of *S. africana* EO in DMSO inhibiting 50% of LOX activity.

### 4.10. Data Analysis

Principal components analysis (PCA) was performed using Xlstat (Adinsoft, Paris, France) [59].

## Figures and Tables

**Figure 1 molecules-24-00204-f001:**
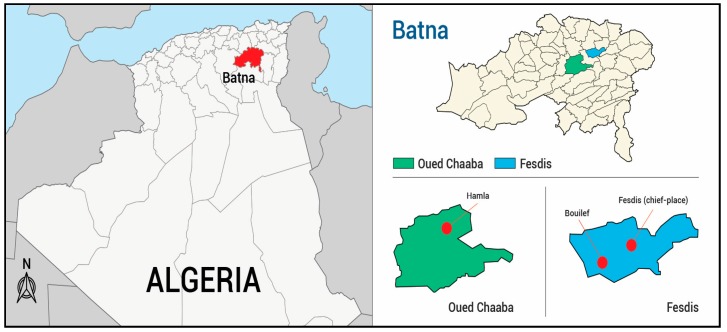
Sampling locations of *Santolina africana* from eastern of Algeria.

**Figure 2 molecules-24-00204-f002:**
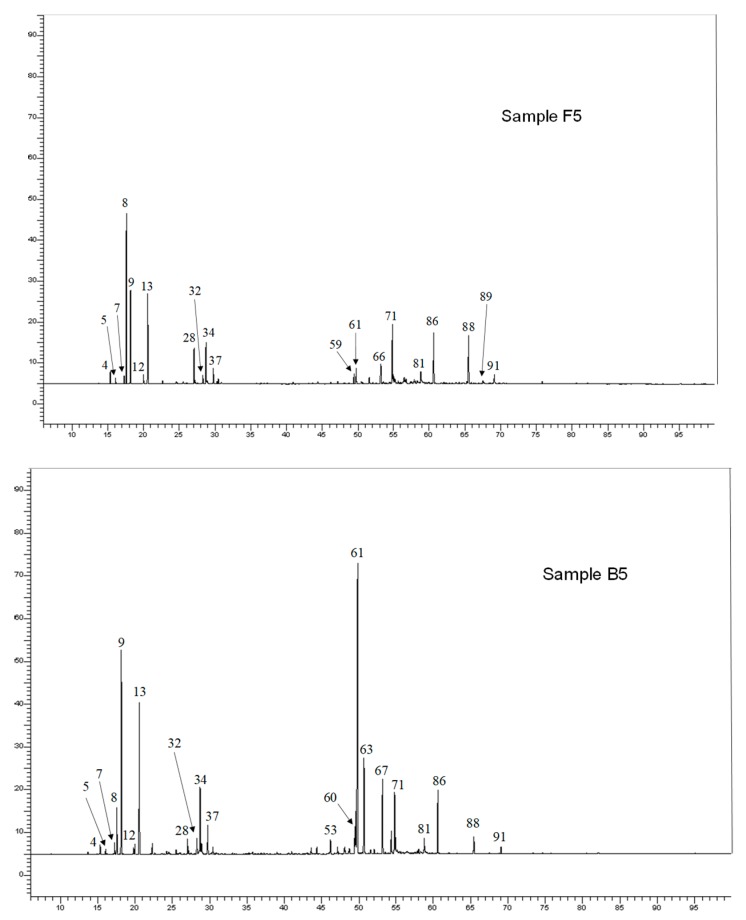
Gas chromatograms of *Santolina africana* EO (samples F5 up and B5 down). The numbered peaks are the identified components (see Appendix A).

**Figure 3 molecules-24-00204-f003:**
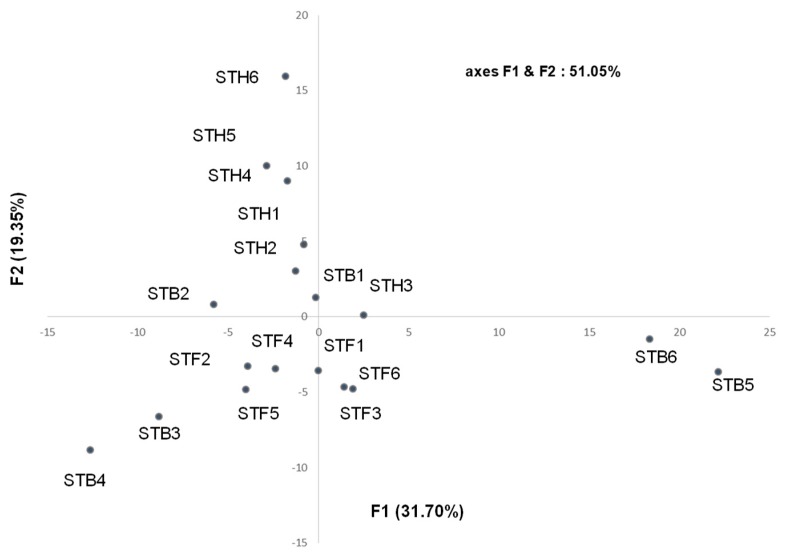
PCA of *Santolina africana* essential oil samples.

**Table 1 molecules-24-00204-t001:** Antimicrobial activity of *S. africana* essential oil.

Microorganisms	Essential Oil (15 μL/disc)	Positive Controls						Negative Control (DMSO)
		CHL	VAN	GMN	CIP	FLU	NY	6.0
*Escherichia coli*	8.00 ± 0.0	25.0 ± 0.0	6.0 ± 0.0	23.0 ± 0.0	35.5 ± 0.7	----	----	6.0
*Klebsiella pneumoniae*	6.0 ± 0.0	21.3 ± 0.6	6.0 ± 0.0	20.0 ± 0.0	30.5 ± 0.7	----	----	6.0
*Staphylococcus aureus*	19.7 ± 0.6	25.5 ± 0.7	17.0 ± 0.0	21.0 ± 0.0	32.0 ± 0.0	----	----	6.0
*Bacillus cereus*	6.0 ± 0.0	29.5 ± 0.7	6.0 ± 0.0	21.0 ± 0.0	37.0 ± 0.0	----	----	6.0
*Candida albicans* ATCC 10231	13.0 ± 0.0	----	----	----	----	6.0 ± 0.0	16.0 ± 0.0	6.0
*Candida albicans* ATCC 26790	15.3 ± 1.5	----	----	----	----	15.0 ± 0.0	19.0 ± 1.0	6.0
*Aspergillus flavus*	20.5 ± 0.7 ^a^	----	----	----	----	6.0 ± 0.0	22.3 ± 0.6	6.0
	13.5 ± 0.7 ^b^	----	----	----	----	6.0 ± 0.0	----	6.0
*Aspergillus fumigatus*	43.0 ± 2.8 ^a^	----	----	----	----	6.0 ± 0.0	33.7 ± 1.2	6.0
	17.5 ± 3.5 ^b^	----	----	----	----	6.0 ± 0.0	-----	6.0
*Fusarium oxysporum*	38.5 ± 2.1 ^a^	----	----	----	----	6.0 ± 0.0	16.0 ± 1.0	6.0
	15.0 ± 0.0 ^b^	----	----	----	----	6.0 ± 0.0	-----	6.0

CHL: Chloramphenicol, VAN: Vancomycin, GMN: Gentamicin, CIP: Ciprofloxacin, FLU: Fluconazole. NY: Nystatin were used as positive controls. Mean values of the growth inhibition zones. in mm. including the disc diameter of 6 mm. ----: Not tested. ^a^: After 3 days. ^b^: After 5 days.

**Table 2 molecules-24-00204-t002:** IC_50_ values and anti-inflammatory activity of *Santolina africana* essential oil.

Activity	Antioxidant		Anti-Inflammatory	
	Essential oil	1.51 ± 0.04	Essential oil	0.065 ± 0.004
	Ascorbic acid	0.02 ± 0.0005	* NDGA	0.013 ± 0.003
Anti-inflammatory activity (percentage inhibition of LOX)				
Concentration ^#^	Inhibition (%)	Concentration ^#^	Inhibition (%)	
0.015	23.4 ± 4.1	0.050	37.5 ± 2.8	
0.025	28.5 ± 5.5	0.075	57.6 ± 3.5	

Values are means of triplicates ± standard deviation; * NDGA: Nordihydroguaiaretic Acid; ^#^ mg/mL.

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
