# Peer review of "Biological Activities and Chemical Composition of Santolina africana Jord. et Fourr. Aerial Part Essential Oil from Algeria: Occurrence of Polyacetylene Derivatives"

_molecules, 2019, doi:10.3390/molecules24010204_

Reviewer 1 Report

The subject matter of the Communication paper – Biological activities and chemical composition of Santolina africana Jord. et Fourr. aerial part essential oil from Algeria: occurrence of the polyacetylenes derivatives [404180] – This article may be published in the MOLECULES journal in the present form.

In this study, the chemical composition of oil samples of Santolina africana isolated from aerial parts at full flowering, were submitted to GC(RI), GC/MS and 13C-NMR analysis. Some components have been found for the first time. The authors have also studied the total antioxidant capacity, anti-inflammatory and antimicrobial activity of the essential oil of S. africana.

This paper is well structured, the materials and methods are quite informative to allow replication of the experiment, the results are clearly presented, the tables and figures are all necessary, complete and clearly presented. The references are adequate. The statistical methods used are correct and adequate.

However, authors should consider the following tips:

1)      On page 7, line 173: “table 4” changes for “table 3”.

2)      On page 7, line 179: “Table 4” change for “Table 3”.

3)      On page 7, Line 203: “Table 3” change for “Table 4”.

Author Response

Reviewer 1

We modify these tips

1) On page 7, line 173: “table 4” changes for “table 3”.

2) On page 7, line 179: “Table 4” changes for “Table 3”.

3) On page 7, Line 203: “Table 3” changes for “Table 4”. We suppressed the table 4. Data of table 3 were included in table 4 (numbered now as Table 3)

Reviewer 2 Report

The authors examined medicinal chemical aspects of a plant, Santolina using diverse experimental techniques. The topic looks interesting and important in the field. Even though the overall scheme in the study seems appropriately designed, the manuscript suffers many drawbacks.

Several statements were made without statistical tests. For example, on line 83,  "...varied drastically from sample to sample" should be evaluated statistically. Another example is the values of IC50s. Several IC50 values were reported in the manuscript without statistical examination.

The materials and methods section was found in need of a significant improvement. For example, for the PCA, it was not clear which matrix (covariance or correlation) was used in the PCA. In addition, the factors examined in the PCA were not described.

The quality of data presentation is not appropriate for publication. For example, in Figure 2, two peaks are expected between peaks 9 and 12, but no peaks were visible. In contrast, several peaks are present between peaks 86 and 88, even though only one peak is expected based on the description in the manuscript.

Several fungi were included in the antimicrobial activity experiment. However, no discussion on the species was included in the manuscript.

I suggest that the authors revise their manuscript responding to comments and resubmit.

Author Response

Reviewer 2

Several statements were made without statistical tests. For example, on line 83,  "...varied drastically from sample to sample" should be evaluated statistically.

Line 83 We modify the sentence.

Another example is the values of IC50s. Several IC50 values were reported in the manuscript without statistical examination.

We check this point in the text.

The materials and methods section was found in need of a significant improvement. For example, for the PCA, it was not clear which matrix (covariance or correlation) was used in the PCA. In addition, the factors examined in the PCA were not described

Line141. We precise the used matrix: covariance. The factors are represented by F1 and F2 (observation axes).

The quality of data presentation is not appropriate for publication. For example, in Figure 2, two peaks are expected between peaks 9 and 12, but no peaks were visible. In contrast, several peaks are present between peaks 86 and 88, even though only one peak is expected based on the description in the manuscript.

Figure 2 shows the gas chromatogram of sample F5 where compounds 10 and 11 were absent. Minor compounds between 86 and 88 are not identified. We added in figure 2 a chromatogram of sample B5.

Several fungi were included in the antimicrobial activity experiment. However, no discussion on the species was included in the manuscript.

We added a sentence concerning the antifungal activity

Reviewer 3 Report

The chemical composition of 18 oil samples of S. africana collected in Algeria was evaluated as well as the antioxidant and antimicrobial activities.

The authors should avoid use: “we get insight, we obtained a series, we constituted one main……..”. Please verify the other ones that are along the manuscript.

There are few examples in which the name of the species is in capital letter (S. Africana), It is necessary to correct it.

It is not adequate to start a sentence with numerals (line 332: 2,5 mL…..)

Lines 96-97: “The composition of S. africana EOs is homogeneous;….”: Maybe it would be preferable to write: The composition of S. africana EOs is generally homogeneous……

It is suggested a Table depicting the main components of the EOs from Algeria already reported. In the map presented in the manuscript the authors should show the regions were the samples of S. africana were already harvested and studied, because such may make easier to the reader verify the harvesting zones.

Although the similarity among sample oils, it is suggested to present the chemical composition of the 18 samples and introduce three chromatograms: one of them showing the general profile of the 16 samples and two chromatograms for samples B5 and B6.

The chemical structures of the polyacetylene compounds reported for the first time in this species should be presented in the manuscript.

Figure 4 can be removed as well as Table 3. Table 4 in the manuscript should be Table 3.

In the essential oil fractionation, the authors should indicate the mass of sample applied on the top of the chromatograph column.

Author Response

Reviewer 3

The authors should avoid use: “we get insight, we obtained a series, we constituted one main……..”. Please verify the other ones that are along the manuscript.

 Lines 108, 124, 132, 146, 174 and 184. We modify several sentences.

There are few examples in which the name of the species is in capital letter (S. Africana), It is necessary to correct it.

 We modify the botanical name.

It is not adequate to start a sentence with numerals (line 332: 2,5 mL…..)

Line 332. We modify the sentence.

Lines 96-97: “The composition of S. africana EOs is homogeneous;….”: Maybe it would be preferable to write: The composition of S. africana EOs is generally homogeneous……

 Line 96-97. We modify the sentence.

It is suggested a Table depicting the main components of the EOs from Algeria already reported. In the map presented in the manuscript the authors should show the regions were the samples of S. africana were already harvested and studied, because such may make easier to the reader verify the harvesting zones.

Two studied samples were collected in Constantine, close to the area of sampling (Batna).

Although the similarity among sample oils, it is suggested to present the chemical composition of the 18 samples and introduce three chromatograms: one of them showing the general profile of the 16 samples and two chromatograms for samples B5 and B6.

We create a supplementary file in order to add this table. The chemical composition of 18 samples was added in Table S1. The GC-FID chromatogram of sample B5 was added in Figure 2.

The chemical structures of the polyacetylene compounds reported for the first time in this species should be presented in the manuscript.

 The chemical structures were added in the supplementary material file (Figure S1).

Figure 4 can be removed as well as Table 3. Table 4 in the manuscript should be Table 3.

Figure 4 was included in the supplementary material file (Figure S2). Table 4 was removed.

In the essential oil fractionation, the authors should indicate the mass of sample applied on the top of the chromatograph column.

The initial amount was present in experimental part (paragraph 4.6, line 301).

Reviewer 4 Report

The reviewed paper concerns the analysis and biological activity of essential oils hydrodistilled from Santolina africana collected in Algeria. After reading this paper I have to say that it was now well thought out. The purpose of this study is unclear. In the introduction the authors show us that the composition of EOs from aerial parts of S. africana exhibited a tremendous chemical variability, and it is because (in authors opinion) that most of the papers report data from one or two EO samples. It is not really true. How about climatic conditions, collection place, seasonal variations, chemotypes….? If your aim was to check the chemical variability you should collect the samples from the different places, not only Batna province.

There is so many other drawbacks and inaccuracies.

My first comment is about polyacetylenes, since the authors mentioned these components in the title. Unfortunately, in the discussion to this paper there is no information why these components were important for the authors? Polyacetylenes are well known components occurring in plants belonging to the Asteraceae family, and were also reported from some Santolina species, e.g. S. corsica or S. rosmarinifolia ssp. rosmarinifolia. On the other hand I am doubtful about the proper identification of these components (paragraph 2.2.). The literature cited: Chanotiya et al. [31] and Sanz et al. [32], does not contain NMR data.

There is also many questions concerning GC analysis.

Table 1. Since the analyses of EO were performed on two different columns, this table should contain the retention indices calculated for both columns. The problem are also percentages. Since these are relative percentages obtained by peak area normalization, without using any correction factor, you cannot mix the percentages of two different columns. The quantification should be done by using different method.

What kind of GC chromatogram is presented on Figure 2 (GC-FID of TIC from GC/MS, which sample)?????

On Figure 2, compounds 12 and 13 are very close, but in the table 1 the RI values are quite different. Are you sure that identification was done in a proper way?

Biological activity

The authors used the agar disc diffusion method for testing the antimicrobial activity of essential oils. It was a very bad choice. This method does not provide true results because only the more water soluble components diffuse into the agar medium, and essential oils are not soluble in water!!!! Other problem is, no standard(s) was/were use

to compare the activity. So, discussion provided in this paper is just a speculation, especially that in the description MIC values from literature data were used, while these were not determined for the examined EOs.

Table 3, presenting anti-inflammatory activity, is really not necessary, since IC50 for this activity is included in table 4.

Why antioxidant activity was measured for EO obtained from sample collected in 2018, while the investigated plant materials were collected in 2016. Are you sure that the composition was same!?

Which EO was studied for antimicrobial activity?

I can not agree for the publication of this paper at its present form.

Author Response

Reviewer 4

The reviewed paper concerns the analysis and biological activity of essential oils hydrodistilled from Santolinaafricana collected in Algeria. After reading this paper I have to say that it was now well thought out. The purpose of this study is unclear. In the introduction the authors show us that the composition of EOs from aerial parts of S. africana exhibited a tremendous chemical variability, and it is because (in authors opinion) that most of the papers report data from one or two EO samples. It is not really true. How about climatic conditions, collection place, seasonal variations, chemotypes….? If your aim was to check the chemical variability you should collect the samples from the different places, not only Batna province.

I agree with these remarks. The literature data are limited (two papers) but the two studied oils exhibited a very different composition. However, the composition exhibiting acenaphtane as major compound create a problem. This identification is probably a mistake. In this work, we choose a limited sampling area to avoid a difference of pedoclimatic conditions or the influence of vegetative stage. The aim of this work is not to demonstrate a chemical variability. For a study of chemical variability, the number of samples and the locations can be elevated. The aim of this work is a chemical characterization in a limited area.

There is so many other drawbacks and inaccuracies.

My first comment is about polyacetylenes, since the authors mentioned these components in the title. Unfortunately, in the discussion to this paper there is no information why these components were important for the authors? Polyacetylenes are well known components occurring in plants belonging to the Asteraceae family, and were also reported from some Santolina species, e.g. S. corsica or S. rosmarinifolia ssp. rosmarinifolia.

If polyacetylene compounds were frequently found in root EOs or extracts from Asteraceae family, this is the first identification of these four compounds simultaneously in aerial parts of EOs.

On the other hand I am doubtful about the proper identification of these components (paragraph 2.2.). The literature cited: Chanotiya et al. [31] and Sanz et al. [32], does not contain NMR data.

Chanotiya et al. [31] contained full spectral data including the carbon-13 NMR data of the two isomers of tonghaosu whereas Sanz et al. [32] described the carbon-13 NMR data of compound 88. The identification of these compounds (paragraph 4.5) was based on comparison of the signals in the carbon-13 NMR spectra of fractions with literature data. DEPT spectra was also recorded to identify CH, CH2 and CH3. This method is frequently developed in the field of phytochemical analysis of essential oils.

There is also many questions concerning GC analysis.

Table 1. Since the analyses of EO were performed on two different columns, this table should contain the retention indices calculated for both columns. The problem are also percentages. Since these are relative percentages obtained by peak area normalization, without using any correction factor, you cannot mix the percentages of two different columns. The quantification should be done by using different method.

The analysis of EOs was performed on two different columns (apolar and polar phases). We calculated the RIs on two columns but only RIs apolar were included in the table. When a coelution occurred on apolar column, it is normal to include the value found on polar column.

Moreover, there is a problem during the pdf creation, the footnote of table 1 was present after the figure 2.

What kind of GC chromatogram is presented on Figure 2 (GC-FID of TIC from GC/MS, which sample)?????

As indicated in the experimental section (4.2), chromatogram presented on Figure 2 was obtained by GC-FID.

On Figure 2, compounds 12 and 13 are very close, but in the table 1 the RI values are quite different. Are you sure that identification was done in a proper way?

Compounds 12 (p-cymene) and 13 (1,8-cineole) were separated by 10 points of RI approximatively 0.6 min. For instance, compound 8 (b-pinene) and 9 (myrcene) were also separated by 10 points (0.6 min).

Biological activity

The authors used the agar disc diffusion method for testing the antimicrobial activity of essential oils. It was a very bad choice. This method does not provide true results because only the more water soluble components diffuse into the agar medium, and essential oils are not soluble in water!!!!

As described in the literature, we used DMSO as solvent with EO to allow the dispersion.

Other problem is, no standard(s) was/were use to compare the activity. So, discussion provided in this paper is just a speculation, especially that in the description MIC values from literature data were used, while these were not determined for the examined EOs.

We added a sentence concerning the comparison of activity (paragraph 2.4).

Table 3, presenting anti-inflammatory activity, is really not necessary, since IC50 for this activity is included in table 4.

We suppressed the table 4.  Data of table 3 were included in table 4 (numbered now as Table 3)

Why antioxidant activity was measured for EO obtained from sample collected in 2018, while the investigated plant materials were collected in 2016. Are you sure that the composition was same!?

Which EO was studied for antimicrobial activity?

Line 329. It’s a mistake. Antioxydant activity was measured with a collective sample collected in 2016 (samples Fesdis F1-F6).

Line 316. Collective sample was used for antimicrobial activity (samples Bouilef and Hamla).

Line 345.  We added “collective sample Fesdis F1-F6”

Round  2

Reviewer 2 Report

Lines 79 to 81, "Yields of EO isolated by hydrodistillation, calculated w/w vs. dry material were higher for plants from Bouilef (0.033-0.166% w/w, samples B1-B6) and Hamla (0.075-0.138% w/w, samples H1-H6) and lesser for plants from Fesdis (0.026-0.114% w/w, samples F1-F6)" needs statistical analysis. Statements like "higher for... and... lesser for plants" need statistical basis. But I could not find it.

The authors include supplementary materials where the complete compositions of oils are available. Therefore, the information in Table 1 is redundant. I suggest the authors to delete Table 1.

In line 143, "two atypical compositions (B5 and B6, Figure 3) were observed", I do not see the statistical basis for "atypical". What is the statistical basis for the statement of "atypical"?

In line 173, it was not shown how the IC50 value of 1.51±0.04 mg/mL was obtained? Was it from a curve fitting? How did the author come up with the value of 1.51 and a statistical uncertainty of 0.04?

Author Response

We took into account all the suggestions and requirements. We revised the manuscript accordingly with all changes highlighted in yellow.

1. Lines 79 to 81, "Yields of EO isolated by hydrodistillation, calculated w/w vs. dry material were higher for plants from Bouilef (0.033-0.166% w/w, samples B1-B6) and Hamla (0.075-0.138% w/w, samples H1-H6) and lesser for plants from Fesdis (0.026-0.114% w/w, samples F1-F6)" needs statistical analysis. Statements like "higher for... and... lesser for plants" need statistical basis. But I could not find it.

I agree to the referee. We added the yields in Table S1 (last line) and modify the sentence: “Yields of EO isolated by hydrodistillation, calculated w/w vs. dry material varied drastically from sample to sample ranging from 0.03 to 0.17% even within a location (Table S1). As it could be seen from Table S1, the highest yields were obtained from Hamla (0.08-0.14%, samples H1-H6) and Bouilef (0.15%, sample B5 and 0.17%, sample B6) and the lower from Bouilef and Fesdis (0.03% for samples B2, B4 and F6)”.

2. The authors include supplementary materials where the complete compositions of oils are available. Therefore, the information in Table 1 is redundant. I suggest the authors to delete Table 1.

We suppressed the table 1. Then, the numbering of tables was modified.

3.In line 143, "two atypical compositions (B5 and B6, Figure 3) were observed", I do not see the statistical basis for "atypical". What is the statistical basis for the statement of "atypical"?

We added the sentence to explain the atypical character of these samples “Indeed, B5 and B6 were discriminated by a high percentage of sesquiterpene hydrocarbons (bicyclogermacrene, (E)-α-bisabolene, γ-curcumene) and particularly germacrene D, 25.3% (B5) and 20.2% (B6) vs. 0-7.5% for the other samples.”

In the figure below (ACP), the discrimination of B5 and B6 samples was due to the group of sesquiterpene hydrocarbons: bicyclogermacrene, (E)-α-bisabolene, γ-curcumene, a-copaene, (E)-b-farnesene (E)-b-caryophyllene and mainly, germacrene D.

4.In line 173, it was not shown how the IC50 value of 1.51±0.04 mg/mL was obtained? Was it from a curve fitting? How did the author come up with the value of 1.51 and a statistical uncertainty of 0.04?

For each concentration of essential oil tested, the graph of inhibition ratio (percent) against increasing concentrations of essential oil was plotted. The assay was carried out in triplicate. IC50 values, which represented the concentration of essential oils that caused 50% neutralization of DPPH radicals, were calculated from the equation of the curve (plot of inhibition percentage against concentration). Results were expressed as mean values of triplicates : IC50 ± Standard Deviation.
